Snake venom NAD glycohydrolases: primary structures, genomic location, and gene structure

Koludarov Ivan ivan.koludarov@oist.jp 1
Aird Steven D. steven.aird@oist.jp 2
1 Ecology and Evolution Unit, Okinawa Institute of Science and Technology , Onna , Kunigami-gun, Okinawa , Japan
2 Ecology and Evolution Unit and Division of Faculty Affairs, Okinawa Institute of Science and Technology , Onna , Kunigami-gun, Okinawa , Japan
Uversky Vladimir
Electronic publication date: 2019 Feb 6
Publication date: 2019
Volume: 7
Electronic Location ID: e6154
Received 2018 Sep 26; Accepted 2018 Nov 25
Copyright: ©2019 Koludarov and Aird
Copyright year: 2019
Copyright holder: Koludarov and Aird
License: This is an open access article distributed under the terms of the Creative Commons Attribution License, which permits unrestricted use, distribution, reproduction and adaptation in any medium and for any purpose provided that it is properly attributed. For attribution, the original author(s), title, publication source (PeerJ) and either DOI or URL of the article must be cited.
License URL: https://creativecommons.org/licenses/by/4.0/

Keywords: Snake venom, NAD glycohydrolase, NADase, CD38, ADP-ribosyl cyclase, Protobothrops mucrosquamatus genome, Exons, Adenosine, Phosphodiesterase, 5′-nucleotidase

Funding: Okinawa Institute of Science and Technology This work was supported by the Okinawa Institute of Science and Technology to the Ecology and Evolution Unit. The funders had no role in study design, data collection and analysis, decision to publish, or preparation of the manuscript.

==============================
NAD glycohydrolase (EC 3.2.2.5) (NADase) sequences have been identified in 10 elapid and crotalid venom gland transcriptomes, eight of which are complete. These sequences show very high homology, but elapid and crotalid sequences also display consistent differences. As in Aplysia kurodai ADP-ribosyl cyclase and vertebrate CD38 genes, snake venom NADase genes comprise eight exons; however, in the Protobothrops mucrosquamatus genome, the sixth exon is sometimes not transcribed, yielding a shortened NADase mRNA that encodes all six disulfide bonds, but an active site that lacks the catalytic glutamate residue. The function of this shortened protein, if expressed, is unknown. While many vertebrate CD38s are multifunctional, liberating both ADP-ribose and small quantities of cyclic ADP-ribose (cADPR), snake venom CD38 homologs are dedicated NADases. They possess the invariant TLEDTL sequence (residues 144–149) that bounds the active site and the catalytic residue, Glu228. In addition, they possess a disulfide bond (Cys121–Cys202) that specifically prevents ADP-ribosyl cyclase activity in combination with Ile224, in lieu of phenylalanine, which is requisite for ADPR cyclases. In concert with venom phosphodiesterase and 5′-nucleotidase and their ecto-enzyme homologs in prey tissues, snake venom NADases comprise part of an envenomation strategy to liberate purine nucleosides, and particularly adenosine, in the prey, promoting prey immobilization via hypotension and paralysis.

Introduction

More than 60 years ago, Bhattacharya (1953) reported that when Bungarus fasciatus venom is incubated with NAD, it releases nicotinamide. This constituted the first evidence that some snake venoms contain an NAD glycohydrolase (NADase) (EC 3.2.2.5). However, like many other non-toxic enzymes, its presence in venoms seemed enigmatic until Aird (2002) proposed that purine nucleosides comprise core elements of the envenomation strategies of most advanced venomous snakes. Adenosine is particularly important because of its hypotensive and neuroprotective (neurosuppressive) activities. Venom NADase augments adenosine release in prey tissues by cleaving β-NAD and NADP to nicotinamide and ADP-ribose, from which adenosine can be liberated by venom and tissue phosphodiesterases in combination with venom and tissue 5′-nucleotidases. Recently, it has been reported that the Deinagkistrodon acutus NADase is capable of hydrolyzing both ATP and ADP to AMP, a function normally supplied by phosphodiesterase (Zhang et al., 2009).

Historical overview of studies on snake venom NADase

Seven years after the Bhattacharya study, Suzuki, Iizuka & Murata (1960) examined 9 Asian snake venoms for NADase activity, using the UV detection method (340 nm) of Zatman, Kaplan & Colowick (1953) and discovered this enzyme in the venoms of Bungarus multicinctus and Trimeresurus gramineus. Venoms reported as negative for NADase included Gloydius blomhoffii, Deinagkistrodon acutus, Ovophis okinavensis, Protobothrops mucrosquamatus, Naja atra, Naja naja, and Ophiophagus hannah.

In a study of 37 elapid, viperid and crotalid venoms, also using UV detection, Tatsuki et al. (1975) confirmed the earlier findings of NADase activity in venoms of the two Bungarus species and further identified it in venoms of Agkistrodon c. contortrix, A. c. mokasen, A. c. laticinctus, A. p. piscivorus, Gloydius blomhoffii, Deinagkistrodon acutus, and Causus rhombeatus, the first viperid examined. All other taxa were reportedly negative for NADase activity (Table 1). Using a succession of five liquid chromatographic procedures they isolated the enzyme from G. blomhoffii venom and characterized it biochemically. The Gloydius enzyme readily hydrolyzed β-NAD and NADP+, and cleaved 3-acetylpyridine adenine dinucleotide, but it did not hydrolyze NADH, NADPH, α-NAD+, or β-nicotinamide mononucleotide (β-NMN) (Tatsuki et al., 1975). They did not estimate the enzyme’s molecular weight.

Table 1 Species negative for NADase activity, according to Tatsuki et al. (1975).

Elapidae	Viperidae	Crotalidae	
Dendroaspis angusticeps	Bitis arietans	Bothrops atrox	
Dendroaspis polylepis	Bitis gabonica	Crotalus adamanteus	
Hemachatus haemachatus	Daboia russellii	Crotalus atrox	
Naja atra	Daboia palestinae	Crotalus basiliscus	
Naja haje	Echis carinatus	Crotalus durissus terrificus	
Naja melanoleuca	Vipera ammodytes	Crotalus viridis viridis	
Naja nivea		Ovophis okinavensis	
Ophiophagus hannah		Protobothrops flavoviridis	
		Protobothrops mucrosquamatus	

Yost & Anderson (1981) characterized the NADase from Bungarus fasciatus venom, and reported that it was a homodimeric glycoprotein of 120–130,000 Da, having a monomeric molecular weight of 62,000 (denaturing SDS PAGE). The enzyme comprised approximately 0.1% of Bungarus fasciatus venom by mass (Yost & Anderson, 1981; Anderson, Yost & Anderson, 1986) while a value of 0.5% was reported from Deinagkistrodon acutus venom (Wu et al., 2002) using a simpler chromatographic procedure with more sophisticated resins.

Huang et al. (1988) investigated the NADase from Deinagkistrodon acutus venom. That enzyme is a homodimeric glycoprotein of about 98,000 Da, having a minimum monomeric molecular mass of 33,500 Da, allowing for a carbohydrate content estimated at 33%. The authors reported that the N-terminal amino acid was proline. As with the Gloydius enzyme (Tatsuki et al., 1975), NADP was the optimal substrate (Huang et al., 1988). The Deinagkistrodon NADase is a metalloenzyme, containing a single, essential copper ion.

Despite these studies, no structural information has been reported for any snake venom NADase. Because our group has completed a series of elapid and crotalid venom gland transcriptomic studies employing high-throughput techniques (Aird et al., 2013; Aird et al., 2015; Aird et al., 2017b), we searched these transcriptomes for the presence of NAD glycohydrolase. It was found in all of them, and we here report their primary structures, possible 3D structures, genomic arrangement, and gene structure.

Materials & Methods

The NADase sequences examined were generated in previously reported studies (Aird et al., 2013; Aird et al., 2015; Aird et al., 2017b). Models of the Micrurus surinamensis and Protobothrops mucrosquamatus NADases and human CD38 were created using GalaxyTBM (Ko et al., 2012) (http://galaxy.seoklab.org/cgi-bin/submit.cgi?type=TBM). Three-dimensional structures were visualized and analyzed using Chimera version 1.13 (Pettersen et al., 2004) (http://www.rbvi.ucsf.edu/chimera). Probable disulfide bonds were created manually based upon cysteine locations and energy minimizations were performed thereafter to optimize the structures. Hydrophilicity/hydrophobicity scores were calculated using the Gravy calculator at: http://www.gravy-calculator.de.

Geneious 8.1.9 (https://www.geneious.com) was used to BLAST elapid and crotalid transcriptomes for NAD glycohydrolase sequences using Gallus gallus CD38 as a query sequence and to align snake venom sequences (Table 2). Genomic scaffolds were examined with ncbi-blast/2.7.1+ suite (https://blast.ncbi.nlm.nih.gov) using Homo sapiens CD38 and Protobothrops mucrosquamatus CD38 as queries and further analyzed in Geneious 8.1.9 to determine exon sequences (Table 2).

Table 2 Transcript IDs and protein accession numbers used in this study.

Taxon	Name	Scaffold ID	Gene ID	Transcript ID	
Homo sapiens	CD38	NC_000004.12	952	NM_001775.3	
Thamnophis sirtalis	CD38	NW_013658259	106543891	XM_014059996.1	
Python bivittatus	CD38	NW_006532481	103059310	XM_007424863.2	
Anolis carolinensis	CD38	NC_014779	100566030	XM_016993323.1	
Xenopus laevis	CD38	NC_030724.1	100036901	NM_001097679.1	
Protobothrops mucrosquamatus	Long CD38	NW_015387543	107292463	XM_015821551.1	
Protobothrops mucrosquamatus	Short CD38	NW_015387543	107292463	XM_015821552.1	
Accession #	
Gallus gallus	CD38	NCBI ADQ89191.1			
Transcript ID	
Micrurus carvalhoi	CD38	DN61384_c0_g1_i1—m.5640			
Micrurus corallinus	CD38	DN100482_c0_g2_i2—m.92			
Micrurus lemniscatus	CD38	DN22889_c0_g1_i1—m.65259			
Micrurus paraensis	CD38	DN86064_c0_g1_i1—m.15110			
Micrurus spixii	CD38	DN121140_c2_g1_i1—m.22327			
Micrurus surinamensis	CD38	DN77054_c0_g1_i1—m.2918			
Ovophis okinavensis	CD38	Oo_comp19518_c0_seq1			
Protobothrops elegans	CD38	Pe_comp350_c0_seq1			
Protobothrops flavoviridis	CD38	Pf_comp3789_c0_seq1			

Results & Discussion

Snake venom NADase amino acid sequences

The NCBI Protein site was searched for vertebrate NAD glycohydrolase sequences and the sequence of chicken ADP-ribosyl cyclase (ADQ89191.1), also known as CD38, was downloaded for use as a query sequence. TBLASTN searches of venom gland transcriptomes of 10 elapid and crotalid species were performed using Geneious 8.1.9. A highly similar sequence was identified in each transcriptome, eight of which were complete (Fig. 1), and NADase transcripts were present at low levels in venom gland transcriptomes of all 30 Protobothrops mucrosquamatus examined by (Aird et al., 2017a). A partial (30-residue), unidentified sequence also occurs in the Ophiophagus hannah genome (L345_15802). The former sequences were aligned with CD38 sequences from Gallus gallus, Xenopus laevis, Anolis carolinensis, and Homo sapiens, using Geneious (Fig. 1).

Figure 1 Alignment of 10 snake venom NADases with human, frog, and chicken CD38 sequences.

The snake NADase sequences lack signal peptides. They are readily distinguished from the former three sequences, and are furthermore clearly resolvable as elapid and crotalid sequences, based upon sequence differences at various positions. Like the Homo sapiens CD38, Micrurus sequences have 12 cysteines, presumably arranged in 6 disulfide bonds (Egea et al., 2012). The two complete Protobothrops sequences have 13 cysteines, like the chicken enzyme, raising the possibility of a covalent homodimer. The truncated C-terminus of the P. flavoviridis sequence is almost certainly a misassembly or less likely, a pseudogene, since the last 14 residues (226–239) and the premature stop codon bear no resemblance to the other sequences. Asterisks denote stop codons. Accession numbers for other vertebrate sequences: Homo sapiens, BAA18966.1; Xenopus laevis, BAL72804.1; Gallus gallus, ADQ89191.1; Anolis carolinensis, XP_016848812.1. Amino acid color scheme: Orange, methionine; Yellow, cysteine; Pink, proline; Gray, aliphatic; Purple, aromatic; Royal Blue, basic; Red, acidic; Green, hydroxylated; Sky Blue, amidated.

None of the venom NADases appears to have a signal peptide, based upon sequence analyses using SignalP 4.1 (Petersen et al., 2011). All are readily distinguished from the former four vertebrate CD38 sequences, as all venom sequences commence with the N-terminal sequence, MPFQNS, rather than with proline, as reported by Huang et al. (1988). Like other vertebrate CD38 sequences, snake venom sequences possess a hydrophilic 14-residue N-terminus (Fig. 1). In all NADases, immediately C-terminal to this hydrophilic block, there is a hydrophobic, 25-amino acid segment containing 17 aliphatic residues (L, V, I, and G), 3 threonine residues, 1 lysine, and 2-3 phenylalanines. These are followed by another 20 residues that are nearly all hydrophilic. The aliphatic segment almost has the appearance of a signal peptide (Fig. 1).

Venom NADases are also clearly resolvable into elapid and crotalid sequences. The Micrurus NADases have 303 amino acids, while crotalid enzymes have 304 and may be readily distinguished based upon sequence differences at various positions. Coralsnake and pitviper sequences display the following respective differences: Q/E52, R or Q/W91, S/R105, H/N145, N/D146, K/N167, D/N170, E/M218, I/T217, N/S219, Q/K224, K/E240, and probably also G/D253, S/N276, I/S278, T/A301, and T/K303, although the Ovophis okinavensis and Protobothrops flavoviridis sequences are incomplete at this point (numbered as in Fig. 1). At several positions, some Micrurus sequences show the crotalid residue while others have a different amino acid.

The M. surinamensis apoprotein has a monomeric molecular weight of 34,206 Da and a predicted pI of 7.93, using the Expasy Compute pI/MW tool. The coralsnake enzymes have masses only slightly higher than the 33,500 predicted for the Deinagkistrodon enzyme (Huang et al., 1988). The Micrurus sequences contain three potential N-glycosylation sites (NKSL, position 125; NGSI, position 214; NRSI, position 274), based upon results obtained using the Expasy NetNGlyc 1.0 Server (http://www.cbs.dtu.dk/services/NetNGlyc/). The first, which is common to elapid and crotalid sequences alike (Fig. 1), has the highest likelihood of being glycosylated, although it may not be since venom NADases lack signal peptides. Nonetheless, Huang et al. (1988) reported a carbohydrate content of about 33% for the Deinagkistrodon NADase, so it seems likely that at least the NKSL at position 125 is N-glycosylated. Based upon analysis with the Expasy NetOGlyc server, there are no probable O-glycosylation sites (Steentoft et al., 2013).

Higher-level structural attributes of vertebrate NADases

Human CD38 is an ecto-enzyme with a long, helical membrane anchor and an intracellular N-terminal segment believed to be a random coil (Malavasi et al., 1992; Prasad et al., 1996; Lee, 2006) (Fig. 2). In addition to its enzymatic activity, CD38 also transduces signals to the cytoplasm. It is thought to regulate metabolism and participates in the pathogenesis of diverse maladies such as inflammation, obesity, diabetes, heart disease, asthma, and aging (Chini et al., 2018). Its enzymatic activity is involved in many of these functions. Moreover, CD38 has been identified as a cell–surface marker in multiple myeloma and other blood-related cancers (Chini et al., 2018).

In contrast, snake venom enzymes are soluble rather than membrane-bound (Tatsuki et al., 1975; Yost & Anderson, 1981), like the Aplysia ADP-ribosyl cyclase (Lee & Aarhus, 1991). While venom NADases could conceivably be membrane-bound in exosomes, they have not been reported as exosomal enzymes (Ogawa et al., 2008), and their elution on Sephadex G-100 is appropriate for soluble enzymes of ∼100 kDa rather than for exosomes (Tatsuki et al., 1975). Exosomal embedding seems further unlikely in that all of the venom NADases reported here possess a very short N-terminal α-helix and a random coil, instead of the long α-helical membrane anchor of human CD38 (Lee, 2006) (Fig. 2A). Snake venom NADase residues 45–302 superimpose almost perfectly upon the crystal structure of the soluble extracellular domain of human CD38, except for the divergent C-termini and the truncated N-termini (3F6Y_A) (Fig. 2B).

Yost & Anderson (1981) found that the Bungarus NADase is a homodimer with a dimeric mass of 125–130 kDa, and a subunit mass of 62 kDa on reducing SDS-PAGE. Huang et al. (1988) reported slightly lower values of 98 kDa and 50 kDa for the Deinagkistrodon enzyme, but apparently employed non-reducing SDS-PAGE, suggesting that the homodimer in that taxon is non-covalent. Interestingly, our Micrurus sequences have 12 cysteine residues, arranged in the 3D structure in a manner consistent with 6 disulfide bonds (C68–C84; C101–C181; C121–C202; C162–C175; C256–C277; C289–C298) (Fig. 2) (Egea et al., 2012), coinciding exactly with the disulfide bonds in CD38 (Fig. 2C); however, the two Protobothrops sequences both have a 13th cysteine in the penultimate position (C-terminus), such that the Protobothrops enzymes could be covalent homodimers (Fig. 1).

The N-terminal 50 residues of the M. surinamensis NADase are slightly hydrophobic, with a Gravy score of 0.142, while the N-terminal 50 residues of human CD38 (BAA18966.1) have a significantly more hydrophobic Gravy score of 0.710. As a result, the soluble venom enzymes appear to have a slightly more compact N-terminal domain than human CD38 (Fig. 2D).

Many invertebrate and vertebrate enzymes exhibiting NADase activity are multifunctional, not only hydrolyzing β-NAD(P)+ to nicotinamide and ADP-ribose, but also exhibiting ADP- and GDP-ribosyl cyclase, and cADPR/cGDPR hydrolase activities (Howard et al., 1993; Lee, Graeff & Walseth, 1997; Ziegler et al., 1997; Ziegler, Jorcke & Schweiger, 1997; Augustin, Muller-Steffner & Schuber, 2000; Ferrero et al., 2014). However, unlike ADP-ribosyl cyclase, human CD38 converts very little β-NAD to cADPR (Lee, 2006).

In contrast to many invertebrate and vertebrate CD38 homologs, Yost & Anderson (1981) found that when β-NAD was hydrolyzed by Bungarus fasciatus NADase, nicotinamide and ADP-ribose were the sole products. The B. fasciatus enzyme does not catalyze the conversion of β-NAD to cyclic ADP-ribose (cADPR). This lack of cyclase activity results in part from the presence of a disulfide bond (C124–C206 in Fig. 1; C121–C202, actual) which is absent in Aplysia ADP-ribosyl cyclase (Tohgo et al., 1994). Moreover, this disulfide bond is present in all snake venom NADases for which we have sequences. Graeff et al. (2009) reported that mutation of Phe221 in Aplysia ADP-ribosyl cyclase (Phe227 in Fig. 1) reduced cADPR production and increased ADPR liberation. Consistent with this conclusion, all snake NADases and at least some vertebrate CD38s have isoleucine in this position (Fig. 1), effectively preventing cADPR formation. Snake venom NADases also all have the conserved TLEDTL (144–149) sequence (Fig. 2A) (residues 149–154 in Fig. 1) that forms the bottom of the active site pocket (Lee, 2006), and the catalytic residue, Glu226 (Glu232 in Fig. 1), which are present in all CD38 molecules (Graeff et al., 2001). Substitution of Glu-146 with Phe, Asn, Gly, Asp, Leu, or Ala resulted in cyclase activity up to 9x higher than of wild-type CD38 (Graeff et al., 2001).

Genome location of vertebrate NAD glycohydrolases/ADP-ribosyl cyclases

We performed genome-wide BLAST searches to locate the NAD glycohydrolase gene in the genomes of Homo sapiens, Gallus gallus, Alligator mississippiensis, Anolis carolinensis, Protobothrops mucrosquamatus, Python bivitattus, Thamnophis sirtalis, and Xenopus laevis. After locating the genes, we manually checked their sequences and compared them with existing annotations of transcriptomic and proteomic data.

In the genomes surveyed, NAD glycohydrolase, CD38, is located in the vicinity of the CC2D2A and PROM1 genes (Ch1L in Xenopus laevis, Ch4 in Homo sapiens and Gallus gallus), usually directly downstream from the FGFBP1 gene. Non-squamate vertebrates have a duplicate gene, called BST1, located upstream from CD38. Squamates apparently lack BST1 in this region, probably due to clade-specific gene loss.

Figure 2 The predicted 3D structure of a soluble venom NADase is more compact than that of its membrane-bound human homolog, owing to its somewhat condensed, soluble N-terminus.

The enzyme from Micrurus surinamensis venom (left) is compared with human CD38 (right). (A) Venom enzymes lack the membrane-spanning alpha-helix and the intracellular domain (residues 1–44) of human CD38. Venom enzymes possess a soluble random coil instead. The two models are shown as ribbon structures with rainbow coloring to distinguish different regions from the N- (blue) to the C-terminus (red). (B) Superimposed portions of the two structures, corresponding to the soluble domain of human CD38, modeled upon the crystal structure of the soluble domain of human CD38 (3F6Y_A). The two structures superimpose almost perfectly except for the divergent C-termini and the truncated N-termini. Residues 45–300 of the M. surinamensis NADase (red) and residues 45–296 of human CD38 (blue) are shown. Left, frontal view; Right, left-side view. (C) Disulfide bonds are identical in the M. surinamensis NADase and human CD38. The Cys121–Cys202 disulfide bond, in combination with other residues, helps to prevent ADP-ribosyl cyclase activity, forcing the conversion of β-NAD and other suitable substrates to ADP-ribose, which is subsequently hydrolyzed to adenosine, the strategic target. (D) The M. surinamensis NADase and human CD38 are both hydrophilic overall, with Gravy scores of −0.292 and −0.306, respectively. However, while the N-terminal 44 residues of the M. surinamensis NADase are slightly hydrophobic (0.142), the N-terminus of CD38 is strongly hydrophobic (0.710). Gravy scores below 0 are more likely globular, hydrophilic proteins, while scores above 0 are more likely membrane-bound and hydrophobic (Magdeldin et al., 2012). Surfaces are rendered to show Kyte-Doolittle hydrophobicity with blue residues being most hydrophilic and red residues being most hydrophobic. Models were created with GalaxyTBM using human CD38 (3F6Y) as a template (Ko et al., 2012) . Disulfide bond formation, energy minimization, and structural manipulations were performed using Chimera 1.13 (Pettersen et al., 2004).

Figure 3 Structure of the NAD glycohydrolase gene from the genome of Protobothrops mucrosquamatus (Aird et al., 2017a), showing locations of the 8 exons, and the amino acid sequence of the enzyme expressed in the venom glands.

The mature venom protein comprises 304 amino acids linked by six disulfide bonds. Exon 1 actually encodes part of the 5′-untranslated region, as well as the N-terminus of the protein, and Exon 8 extends well beyond the stop codon, but only the mature protein sequence is shown here. Because of split arginine codons that bridge the Exon 6–7 and 7–8 boundaries, splicing out the sixth exon in the short P. mucrosquamatus NADase variants would retain the arginine after the splice. Distances between exons shown above are for purposes of illustration only, and are not proportionally scaled.

Figure 4 In Protobothrops mucrosquamatus, the 8-exon gene is transcribed into a long NADase and a shorter form that lacks exon 6.

(A) Its deletion results in the loss of 31 amino acids from Ser223-Gln253. With split arginine codons at both the exon 6–7 and exon 7–8 boundaries, no amino acid substitution occurs in the shortened (272-residue) structure. (B) The effect of removing these 31 residues is to delete the loose helical region (Ser223-Gln253) from the center of the molecule, including the catalytic residue, Glu229, (Glu232 in Fig. 1) leaving the remainder of the structure essentially intact. If this alternately spliced protein is expressed, it is difficult to imagine what its function might be. Models were created using GalaxyTBM (Ko et al., 2012). Disulfide bond formation, energy minimization, and structural manipulations were performed using Chimera 1.13 (Pettersen et al., 2004). Amino acid classes are colored as in Fig. 1.

Gene structure of vertebrate NAD glycohydrolases/ADP-ribosyl cyclases

Human CD38 displays similar intron-exon architecture to that seen in the invertebrate, Aplysia kurodai ADP-ribosyl cyclase (Nata et al., 1995), suggesting that this architecture is highly conserved. It comprises eight exons, extending more than 77 kb in the human genome (Nata et al., 1997). However, in Protobothrops mucrosquamatus (and presumably in other venomous snakes) the 8-exon gene is transcribed in two variants, a long NADase similar to CD38 of other vertebrates (Fig. 3), and a shorter form that lacks exon 6 (Fig. 4). Exon 6 contains no cysteines, and its deletion results in the loss of 31 amino acids from Ser223-Gln253 (Fig. 4A). Owing to the presence of split arginine codons at both the exon 6–7 and exon 7–8 boundaries, no amino acid substitution occurs in the shortened (272-residue) structure. Instead, the effect of removing these 31 residues is to delete the loose helical region (Ser223-Gln253) from the center of the molecule, including the essential catalytic residue, Glu229 (Fig. 4A; Glu232 in Fig. 1), while leaving the remainder of the structure essentially intact (Fig. 4B). If this protein is expressed, it is difficult to imagine what its function might be, but with its precise alternate splicing, this does not seem like a pseudogene.

Conclusions

All snake venom gland transcriptomes and venomous snake genomes published to date contain sequences for NAD glycohydrolases, pointing to a significant role in envenomation. At least some crotalids may produce two forms of this enzyme. Strategically, the function of snake venom NADases is to drive the release of adenosine from NADP and β-NAD in prey tissues and to block its conversion to cADPR. Guanosine is also released from NGD. Both purines contribute to prey immobilization via hypotension/circulatory shock and paralysis caused by neurosuppression (Aird, 2002; Aird, 2009).

Supplemental Information

Supplemental Information 1 Venom gland transcriptomic sequences employed in this study

Click here for additional data file.

We thank Alexander S. Mikheyev for critical reading of the manuscript that substantially improved its organization.

Additional Information and Declarations

Competing Interests

Author Contributions

Data Availability

The authors declare there are no competing interests.

Ivan Koludarov and Steven D. Aird conceived and designed the experiments, performed the experiments, analyzed the data, prepared figures and/or tables, authored or reviewed drafts of the paper, approved the final draft.

The following information was supplied regarding data availability:

The raw data is included in Fig. 1.

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
