# Peer review of "Snake venom NAD glycohydrolases: primary structures, genomic location, and gene structure"

_PeerJ, doi:10.7717/peerj.6154_

## Round 0.1 · original submission · Major Revisions

Please address all the critical issues raised by the reviewers and revise your manuscript accordingly.

Reviewer 1 ·

Basic reporting

The manuscript is well organized and well written. The authors familiar with the progress in the field and provide sufficient background.

Experimental design

In Figure2:
The authors compare the structure of soluble venom NADase with human homolog CD38 using predicted 3D structure of two proteins. Actually, the structure of human CD38 extracellular domain has been solved in 2005 already (PDB ID: 1YH3). Why authors using the predicted structure to do analysis? It is more reasonable to analysis the structure based on a solved structure than predicted one.

Validity of the findings

The title of Figure2 is “The predicted 3D structure of a soluble venom NADase is more compact than its membrane-bound human homolog.”
The title of Figure4 is “Predicted 3D structures of short (left) and long (right) NADases from Protobothrops mucrosquamatus venom.”
Why the panel A of figure2 and figure4 looks exactly same?
It is unreliable that the human CD38 has exactly same structure with Protobothrops mucrosquamatus venom.

Additional comments

In this manuscript, the authors searched 10 elapid and crotalid species’ venom gland transcriptomes and reported all snake venom gland transcriptomes contain sequences of NAD glycohydrolases. The authors found the key residues requisite for NADase activity are conserved in snake venom. In addition, structural analysis revealed that a disulfide bond that related with cyclase activity is present in all snake venom NADases for which they have sequences. This manifest the Aird’s group earlier snake envenomation hypothesis: “purine nucleosides comprise core elements of the envenomation strategies of most advanced venomous snakes”. The results may provide new clues for the community to study the NADase activity and envenomation strategies of venomous snakes

The authors describe the detailed sequence difference on line 130 and 131. Mention the residue number is meaningless unless coordinate it with the protein structure. It is better if the authors could label these residues on the structure and the discuss the potential influence of these residues on the function.

The authors mentioned: “Snake venom NADases also all have Glu146 (Glu151 in
191 Figure 1), which is essential for NADase activity and is present in all CD38 molecules.” Why Glu146 is important for NADase activity? It is helpful if author could label the residue on the structure. Similarly, where is the 31 amino acids (delete in short form) locate, what's the possible function of this part?

Reviewer 2 ·

Basic reporting

No comment, see "General Comments".

Experimental design

No comment, see "General Comments".

Validity of the findings

No comment, see "General Comments".

Additional comments

1. Although it’s appreciated that snake venom enzymes are soluble instead of membrane-bound, it is still not obvious to tell from the Kyte-Doolittle hydrophobicity surface plots shown in Fig. 2B that the snake protein is more hydrophilic than the human homolog. It would be more helpful if the authors can provide an overall hydrophobicity score (e.g., GRAVY value) of each protein and see if the snake protein (at least in the N-terminal region) is more hydrophilic compared to the human homolog.
2. In Figure 1, it would be better to explain color coding scheme. Line 159, “venom NADase could be membrane-bound in exosomes”, does this statement have literature support? Line 131, “G/D254” should be “G/D253”; also what about I/T217 and N/S219? Line 154, “in” should be “is”.
3. It would be helpful if the authors can annotate the predicted disulfide bonds on the Micrurus NADase model shown in Fig. 2A.
4. To support their claim in Line 166 “venom enzymes apparently have a slightly more compact shape than human CD38”, it would be better if the authors can provide some numbers characterizing the overall shape of the protein, for example, distances between conserved amino acid residues, and annotate on their models.
5. It would be more informative if the authors can point out in the introduction challenges obtaining structures for snake venom NADases.

---

## Round 0.2 · accepted · Accept

All critical issues raised by the reviewers were appropriately addressed and the manuscript was revised accordingly. Therefore, the revised version can be accepted in its current form.

# Reviewer 1 ·

Basic reporting

No comment.

Experimental design

No comment.

Validity of the findings

No comment.

Additional comments

I have gone over the revised manuscript and the comments, and I am happy with this version. I recommend publication.

Reviewer 2 ·

Basic reporting

See General Comments.

Experimental design

See General Comments.

Validity of the findings

See General Comments.

Additional comments

In the revised manuscript, the authors nicely addressed the previous reviewers' comments. GRAVY scores were nicely incorporated into the manuscript and supported the authors' claim. Their efforts to modify the wording and figure legends are also appreciated. Overall, this revised version is more informative and provide more evidence to back up the findings.